# COPY THAT! EDITING SEQUENCES BY COPYING SPANS

## ABSTRACT

Neural sequence-to-sequence models are finding increasing use in editing of documents, for example in correcting a text document or repairing source code. In this paper, we argue that existing seq2seq models (with a facility to copy single tokens) are *not* a natural fit for such tasks, as they have to explicitly copy each unchanged token. We present an extension of seq2seq models capable of copying entire spans of the input to the output in one step, greatly reducing the number of decisions required during inference. This extension means that there are now many ways of generating the same output, which we handle by deriving a new objective for training and a variation of beam search for inference that explicitly handle this problem.

In our experiments on a range of editing tasks of natural language and source code, we show that our new model consistently outperforms simpler baselines.

## 1 INTRODUCTION

Intelligent systems that *assist* users in achieving their goals have become a focus of recent research. One class of such systems are intelligent editors that identify and correct errors in documents while they are written. Such systems are usually built on the seq2seq (Sutskever et al., 2014) framework, in which an input sequence (the current state of the document) is first encoded into a vector representation and a decoder then constructs a new sequence from this information. Many applications of the seq2seq framework require the decoder to copy some words in the input. An example is machine translation, in which most words are generated in the target language, but rare elements such as names are copied from the input. This can be implemented in an elegant manner by equipping the decoder with a facility that can "point" to words from the input, which are then copied into the output (Vinyals et al., 2015; Grave et al., 2017; Gulcehre et al., 2016; Merity et al., 2017).

*Editing* sequences poses a different problem from other seq2seq tasks, as in many cases, *most* of the input remains unchanged and needs to be reproduced. When using existing decoders, this requires painstaking word-by-word copying of the input. In this paper, we propose to extend a decoder with a facility to copy entire spans of the input to the output in a single step, thus greatly reducing the number of decoder steps required to generate an output. This is illustrated in Figure 1, where we show how our model inserts two new words into a sentence by copying two spans of (more than) twenty tokens each.

However, this decoder extension exacerbates a well-known problem in training decoders with a copying facility: a target sequence can be generated in many different ways when an output token can be generated by different means. In our setting, a sequence of tokens can be copied token-by-token, in pairs of tokens, ..., or in just a single step. In practice, we are interested in encouraging our decoder to use as few steps as possible, both to speed up decoding at inference time as well as to reduce the potential for making mistakes. To this end, we derive a training objective that marginalises over all different generation sequences yielding the correct output, which implicitly encourages copying longer spans. At inference time, we solve this problem by a variation of beam search that "merges" rays in the beam that generate the same output by different means.

In summary, this paper (i) introduces a new sequence decoder able to copy entire spans (Sect. 2); (ii) derives a training objective that encourages our new decoder to copy *long* spans when possible, as well as an adapted beam search method approximating the exact objective; (iii) includes extensive experiments showing that the span-copying decoder improves on editing tasks on natural language and program source code (Sect. 4).

**Input**

charles ▷dea lt ry ◁ ▷loc ock ◁ ( september 27 , 1862 – may 13 , 1946 ) , was a british literary scholar , who wrote on a wide array of subjects , including chess , billiards and ▷cro que t ◁ .

$a_1$: Copy$(1:28)$

$a_4$: Copy$(28:48)$

**Output**

charles ▷dea lt ry ◁ ▷loc ock ◁ ( september 27 , 1862 – may 13 , 1946 ) , was a british literary scholar and translator , who wrote on a wide array of subjects , including chess , billiards and ▷cro que t ◁ .

Figure 1: Sample edit generated by our span-copying model on the WikiAtomicEdits dataset on the edit representation task of Yin et al. (2019). ▷ and ◁ represent the BPE start/end tokens. The model first copies a long initial span of text Copy$(1:28)$. The next two actions generate the tokens "and" and "translator", as in a standard sequence generation models. Then, the model again copies a long span of text and finally generates the end-of-sentence token (not shown).

## 2 MODEL

The core of our new decoder is a span-copying mechanism that can be viewed as a generalisation of pointer networks used for copying single tokens (Vinyals et al., 2015; Grave et al., 2017; Gulcehre et al., 2016; Merity et al., 2017). Concretely, modern sequence decoders treat copying from the input sequence as an alternative to generating a token from the decoder vocabulary, *i.e.* at each step, the decoder can either generate a token $t$ from its vocabulary or it can copy the $i$-th token of the input. We view these as potential *actions* the decoder can perform and denote them by Gen$(t)$ and Copy$(i)$.

Formally, given an input sequence $\boldsymbol{in} = in_1 \ldots in_n$, the probability of a target sequence $o_1 \ldots o_m$ is commonly factorised into sequentially generating all tokens of the output.

$$p(o_1 \ldots o_m \mid \boldsymbol{in}) = \prod_{1 \leq j \leq m} p(o_j \mid \boldsymbol{in}, o_1 \ldots o_{j-1}) \tag{1}$$

Here, $p(o_j \mid \boldsymbol{in}, o_1 \ldots o_{j-1})$ denotes the probability of generating the token $o_j$, which is simply the probability of the Gen$(t)$ action in the absence of a copying mechanism.[1] When we can additionally copy tokens from the input, this probability is the sum of probabilities of all correct actions. To formalise this, we denote evaluation of an action $a$ into a concrete token as $[\![a]\!]$, where $[\![\text{Gen}(t)]\!] = t$ and $[\![\text{Copy}(i)]\!] = in_i$. Using $q(a \mid \boldsymbol{o})$ to denote the probability of emitting an action $a$ after generating the partial output $\boldsymbol{o}$, we complete Eq. (1) by defining

$$p(o_j \mid o_1 \ldots o_{j-1}) = \sum_{a, [\![a]\!]=o_j} q(a \mid o_1 \ldots o_{j-1}),$$

*i.e.* the sum of the probabilities of all correct actions.

**Modelling Span Copying** In this work, we are interested in copying whole subsequences of the input, introducing a sequence copying action Copy$(i:j)$ with $[\![\text{Copy}(i:j)]\!] = in_i \ldots in_{j-1}$ (indexing follows the Python $\boldsymbol{in}\,[i:j]$ notation here). This creates a problem because the number of actions required to generate an output token sequence is not equal to the length of the output sequence anymore; indeed, there may be many action sequences of different length that can produce the correct output.

As an example, consider Fig. 2, which illustrates all action sequences generating the output $a\,b\,f\,d\,e$ given the input $a\,b\,c\,d\,e$. For example, we can initially generate the token $a$, or copy it from the input, or copy the first two tokens. If we chose one of the first two actions, we then have the choice of either generating the token $b$ or copying it from the input and then have to generate the token $f$. Alternatively, if we initially choose to copy the first two tokens, we have to generate the token $f$ next. We can compute the probability of generating the target sequence by traversing the diagram from the right to the left. $p(\epsilon \mid a\,b\,f\,d\,e)$ is simply the probability of emitting a stop token and requires no recursion. $p(e \mid a\,b\,f\,d)$ is the sum of the probabilities $q(\text{Gen}(e) \mid a\,b\,f\,d) \cdot p(\epsilon \mid a\,b\,f\,d\,e)$ and $q(\text{Copy}(4:5) \mid a\,b\,f\,d) \cdot p(\epsilon \mid a\,b\,f\,d\,e)$, which re-uses the term we already computed. Following this strategy, we can compute the probability of generating the output token sequence by computing

---

[1] Note that all occurrences of $p$ (and $q$ below) are implicitly (also) conditioned on the input sequence $\boldsymbol{in}$, and so we drop this in the following to improve readability.

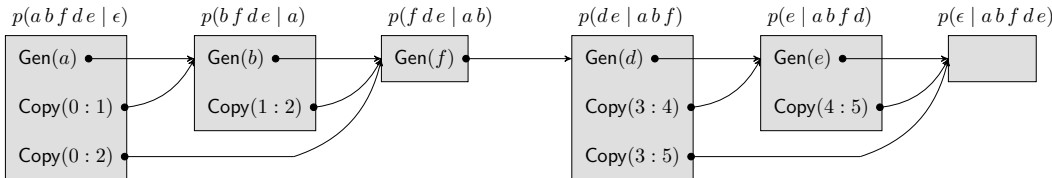

Figure 2: Illustration of different ways of generating the sequence $a\,b\,f\,d\,e$ given an input of $a\,b\,c\,d\,e$. Each box lists all correct actions at a given point in the generation process, and the edges after an action indicate which suffix token sequence still needs to be generated after it. We use $\epsilon$ to denote the empty sequence, either as prefix or suffix.

probabilities of increasing longer suffixes of it (essentially traversing the diagram in Fig. 2 from right to left).

Formally, we reformulate Eq. (1) into a recursive definition that marginalises over all different sequences of actions generating the correct output sequence, following the strategy illustrated in Fig. 2. For this we define $|a|$, the length of the output of an action, *i.e.*, $|\mathsf{Gen}(t)| = 1$ and $|\mathsf{Copy}(i : j)| = j - i$. Note that we simply assume that actions $\mathsf{Copy}(i : j)$ with $j \leq i$ do not exist.

$$p(o_{k+1} \ldots o_m \mid o_1 \ldots o_k) = \sum_{\substack{a, \exists \ell. |a| = \ell \\ [\![a]\!] = o_{k+1} \ldots o_{k+\ell}}} q(a \mid o_1 \ldots o_k) \cdot p(o_{k+\ell+1} \ldots o_m \mid o_1 \ldots o_{k+\ell}) \quad (2)$$

Note that here, the probability of generating the correct suffix is only conditioned on the sequence generated so far and *not* on the concrete actions that yielded it. In practice, we implement this by conditioning our modelling of $q$ at timestep $k$ on a representation $\boldsymbol{h}_k$ computed from the partial output sequence $o_1 \ldots o_k$. In RNNs, this is modelled by feeding the sequence of emitted tokens into the decoder, no matter how the decoder determined to emit these, and thus, one $\mathsf{Copy}(i : j)$ action may cause the decoder RNN to take multiple timesteps to process the copied token sequence. In causal self-attentional settings, this is simply the default behaviour.

We found that using the marginalisation in Eq. (2) during training is crucial for good results. In initial experiments, we tried an ablation in which we generate a per-token loss based only on the correct actions at each output token, without taking the remainder of the sequence into account (*i.e.*, at each point in time, we used a "multi-hot" objective in which the loss encourages picking any one of the correct actions). In this setting, training yielded a decoder which would most often only copy sequences of length one, as the objective was not penalising the choice of long action sequences explicitly. Our marginalised objective in Eq. (2) does exactly that, as it explicitly reflects the cost of having to emit more actions than necessary, pushing the model towards copying longer subsequences. Finally, note that for numerical stability purposes our implementation works on the log-probability space as it is common for such methods, implementing the summation of probabilities with the standard log-sum-exp tricks.

**Modelling Action Choices**   It remains to explain how we model the per-step action distribution $q(a \mid \boldsymbol{o})$. We assume that we have per-token encoder representations $\boldsymbol{r}_1 \ldots \boldsymbol{r}_n$ of all input tokens and a decoder state $\boldsymbol{h}_k$ obtained after emitting the prefix $o_1 \ldots o_{k-1}$. This can be the state of an RNN cell after processing the sequence $o_1 \ldots o_k$ (potentially with attention over the input) or the representation of a self-attentional model processing that sequence.

As in standard sequence decoders, we use an output embedding projection applied to $\boldsymbol{h}_k$ to obtain scores $s_{k,v}$ for all tokens in the decoder vocabulary. To compute a score for a $\mathsf{Copy}(i : j)$ action, we use a linear layer $\boldsymbol{W}$ to project the concatenation $\boldsymbol{r}_i \| \boldsymbol{r}_j$ of the (contextualised) embeddings of the respective input tokens to the same dimension as $\boldsymbol{h}_k$ and then compute their inner product:

$$s_{k,(i,j)} = (\boldsymbol{W} \cdot (\boldsymbol{r}_i \| \boldsymbol{r}_j)) \cdot \boldsymbol{h}_k^\top$$

We then concatenate all $s_{k,v}$ and $s_{k,(i,j)}$ and apply a softmax to obtain our action distribution $q(a \mid \boldsymbol{o})$. Note that for efficient computation in GPUs, we compute the $s_{k,(i,j)}$ for all $i$ and $j$ and mask all entries where $j \leq i$.

---

**Algorithm 1** Python-like pseudocode of beam search for span-copying decoders.

---

```
def beam_search(beam_size)
  beam = [ {toks: [START_OF_SEQ], prob: 1} ]
  out_length = 1
  while unfinished_rays(beam):
    new_rays = []
    for ray in beam:
      if len(ray.toks) > out_length or ray.toks[-1] == END_OF_SEQ:
        new_rays.append(ray)
      else:
        for (act, act_prob) in q(·| ray.toks):
          new_rays.append({toks: ray.toks ‖ ⟦act⟧, prob: ray.prob*act_prob})
    beam = top_k(group_by_toks(new_rays), k=beam_size)
    out_length += 1
  return beam
```

---

**Training Objective**   We train in the standard supervised sequence decoding setting, feeding to the decoder the correct output sequence independent of its decisions. We train by maximising $p(\boldsymbol{o} \mid \epsilon)$ unrolled according to Eq. (2). One special case to note is that we make a minor but important modification to handle generation of out-of-vocabulary words: *iff* the correct token can be copied from the input, Gen(UNK) is considered to be an incorrect action; otherwise only Gen(UNK) is correct. This is necessary to avoid pathological cases in which there is no action sequence to generate the target sequence correctly.

**Beam Decoding**   Our approach to efficiently evaluate Eq. (2) at training time relies on knowledge of the ground truth sequence and so we need to employ another approach at inference time. We use a variation of standard beam search which handles the fact that action sequences of the same length can lead to sequences of different lengths. For this, we consider a forward version of Eq. (2) in which we assume to have a set of action sequences $\mathcal{A}$ and compute a lower bound on the true probability of a sequence $o_1 \ldots o_k$ by considering all action sequences in $\mathcal{A}$ that evaluate to $o_1 \ldots o_k$:

$$p(o_1 \ldots o_k) \geq \sum_{\substack{[a_1 \ldots a_n] \in \mathcal{A} \\ \llbracket a_1 \rrbracket \| \ldots \| \llbracket a_n \rrbracket = o_1 \ldots o_k}} \prod_{1 \leq i \leq n} q(a_i \mid \llbracket a_1 \rrbracket \| \ldots \| \llbracket a_{i-1} \rrbracket). \tag{3}$$

If $\mathcal{A}$ contains the set of all action sequences generating the output sequence $o_1 \ldots o_k$, Eq. (3) is an equality. At inference time, we under-approximate $\mathcal{A}$ by generating likely action sequences using beam search. However, we have to explicitly implement the summation of the probabilities of action sequences yielding the same output sequence. This could be achieved by a final post-processing step (as in Eq. (3)), but we found that it is more effective to "merge" rays generating the same sequence during the search. In the example shown in Fig. 2, this means to sum up the probabilities of (for example) the action sequences Gen($a$)Gen($b$) and Copy(0 : 2), as they generate the same output. To achieve this grouping of action sequences of different lengths, our search procedure is explicitly considering the length of the generated token sequence and "pauses" the expansion of action sequences that have generated longer outputs. We show the pseudocode for this procedure in Alg. 1, where merging of different rays generating the same output is done using `group_by_toks`.

## 3   RELATED WORK

Copying mechanisms are common in neural natural language processing. Starting from pointer networks (Vinyals et al., 2015), such mechanisms have been used across a variety of domains (Allamanis et al., 2016; Gu et al., 2016; See et al., 2017) as a way to copy elements from the input to the output, usually as a way to alleviate issues around rare, out-of-vocabulary tokens such as names. Marginalizing over a single token-copying action and a generation action has been previously considered (Allamanis et al., 2016; Ling et al., 2016) but these works do not consider spans longer than one "unit".

Zhou et al. (2018) proposes a method to copy spans (for text summarization tasks) by predicting the start and end of a span to copy. However, it lacks a marginalization over different actions generating

the sequences and instead uses teacher forcing towards the longest copyable span; and it does not adapt its inference strategy to reflect this marginalization.

Our method is somewhat related to the work of van Merriënboer et al. (2017); Grave et al. (2019), who consider "multiscale" generation of sequences using a vocabulary of potentially overlapping word fragments. Doing this also requires to marginalise out different decoder actions that yield the same output: in their case, generating a sequence from different combinations of word fragments, in contrast to our problem of generating a sequence token-by-token or copying a span. Hence, their training objective is similar to our objective in Eq. (2). A more important difference is that they use a standard autoregressive decoder in which the emitted word fragments are fed back as inputs. This creates the problem of having different decoder states for the same output sequence (dependent on its decomposition into word fragments), which van Merriënboer et al. (2017) resolve by averaging the states of the decoder (an RNN using LSTM cells). Instead, we are following the idea of the graph generation strategy of Liu et al. (2018), where a graph decoder is only conditioned on the partial graph that is being extended, and not the sequence of actions that generated the graph.

Recently, a number of approaches to sequence generation avoiding the left-to-right paradigm have been proposed (Welleck et al., 2019; Stern et al., 2019; Gu et al., 2019; Lee et al., 2018), usually by considering the sequence generation problem as an iterative refinement procedure that changes or extends a full sequence in each iteration step. Editing tasks could be handled by such models by learning to refine the input sequence with the goal of generating the output sequence. However, besides early experiments by Gu et al. (2019), we are not aware of any work that is trying to do this. Note however that our proposed span-copying mechanism is also naturally applicable in settings that require duplication of parts of the input, *e.g.* when phrases or subexpressions need to be appear several times in the output (*cf.* `obj` in Figure 3 for a simple example). Finally, sequence-refinement models could also be extended to take advantage of our technique without large modifications, though we believe the marginalisation over all possible insertion actions (as in Eq. (2)) to be intractable in this setting. Similarly, Grangier & Auli (2017) present QuickEdit, a machine translation method that accepts a source sentence (*e.g.* in German) a guess sentence (*e.g.* in English) that is annotated (by humans) with change markers. It then aims to improve upon the guess by generating a better target sentence avoiding the marked tokens. This is markedly different as the model accepts as input the spans that need to be removed or retained in the guess sentence. In contrast, our model needs to automatically infer this information.

An alternative to sequence generation models for edits is the work of Gupta et al. (2017), who propose to repair source code by first pointing to a single line in the output and then only generate a new version of that line. However, this requires a domain-specific segmentation of the input – lines are often a good choice for programs, but (multi-line) statements or expressions are valid choices as well. Furthermore, the approach still requires to generate a sequence that is similar to the input line and thus could profit from our span-copying approach.

## 4 EXPERIMENTAL EVALUATION

We have implemented our span-copying decoder in PyTorch on top of both RNNs and Transformer models and will release the source code on `http://shown/after/double/blind`. We evaluate the our RNN-based implementation on two types of tasks. First, we consider correction-style tasks in which a model has to identify an error in an input sequence and then generate an output sequence that is a corrected version of the input. Second, we evaluate the performance of our models in the more complex setting of learning edit representations (Yin et al., 2019). In the evaluation below, COPY+SEQ2SEQ denotes a variant of our SEQCOPYSPAN in which the decoder can only copy single tokens (*i.e.*, it is a biGRU encoder with a GRU decoder with token copying).

### 4.1 CORRECTION TASKS

Correction tasks were one of the core motivations for our new decoding strategy, as they usually require to reproduce most of the input without changing it, whereas only few tokens are removed, updated or added. We consider both corrections on source code as well as on natural language.

Table 1: Evaluation of models on the code repair task. Given an input code snippet, each model needs to predict a corrected version of that code snippet. "Structural Match" indicates that the generated output is identical to the target output up to renaming the identifiers (*i.e.*, variables, functions).

|  | Accuracy | Accuracy@20 | MRR | Structural Match |
|---|---|---|---|---|
| On BFP$_{small}$ | | | | |
| Tufano et al. (2019) | 9.2% | 43.5% | — | — |
| COPY+SEQ2SEQ | 14.8% | 42.0% | 0.177 | 18.2% |
| SEQCOPYSPAN | **17.7%** | **45.0%** | **0.247** | **21.2%** |
| SEQCOPYSPAN (greedy) | 15.3% | — | — | 17.9% |
| SEQCOPYSPAN (merge rays at end) | 17.5% | 41.6% | 0.242 | **21.2%** |
| SEQCOPYSPAN (force copy longest) | 14.2% | 33.7% | 0.174 | 14.2% |
| On BFP$_{medium}$ | | | | |
| Tufano et al. (2019) | 3.2% | 22.2% | — | — |
| COPY+SEQ2SEQ | 7.0% | 23.8% | 0.073 | 9.4% |
| SEQCOPYSPAN | **8.0%** | **25.4%** | **0.105** | **13.7%** |

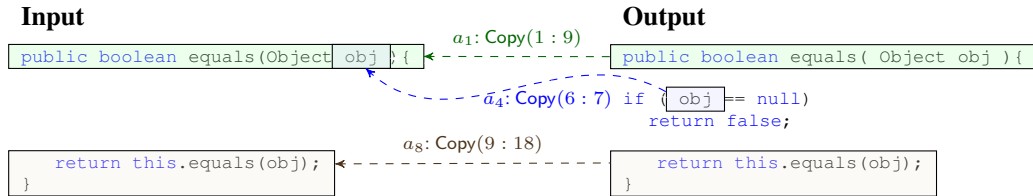

Figure 3: Generation of a test example in BFP$_{small}$ (slightly modified for space and clarity). The SEQCOPYSPAN model learns to copy long spans while generating the necessary edits. The non-highlighted tokens in the output are generated using $\mathsf{Gen}(t)$, whereas all other tokens are copied from the input.

**Code Repair** Automated code repair systems (Monperrus, 2018) are commonly composed of two components, namely a (heuristic) component that suggests potentially fixed versions of the input, and an oracle (*e.g.*, a harness executing a test suite) that checks the candidates for correctness. Recent software engineering research has started to implement the heuristic component using seq2seq models (Chen et al., 2018; Tufano et al., 2019; Lutellier et al., 2019). The models are usually viewed as language models (conditioned on the faulty code) or directly employ standard neural machine translation pipelines mapping from "faulty" to "correct" code. The task usually only requires minor changes to the input code and consequently most of the input is copied into the output. We believe that our model is a natural match for this setting.

To test this hypothesis, we use the two bug-fix pair (BFP) datasets of Tufano et al. (2019). The BFP$_{small}$ dataset contains pairs where each snippet has at most 50 tokens and the BFP$_{medium}$ dataset has Java snippets containing from 50 up to 150 tokens. The original version of the code has some form of a bug, whereas the edited version fixes the relevant bug. This corpus was constructed by scraping Git commits and filtering for those that the commit message suggests that the edit fixes a bugs. For both the COPY+SEQ2SEQ and SEQCOPYSPAN models we employ a 2-layer biGRU as an encoder and a single layer GRU decoder. The hidden dimensions of the GRUs are 128, whereas the embedding layer has a dimensionality of 32. Note that the vocabulary size for this task is just 400 by construction of the dataset. We employ a Luong-style (Luong et al., 2015) attention mechanism in the the decoders of both models.

Table 1 shows the results of our models, as well as the original results reported by Tufano et al. (2019). Overall, the SEQCOPYSPAN model performs better on both datasets, achieving better prediction accuracy. This suggests that the span-copying mechanism is indeed beneficial in this setting, as it becomes quite visible in a qualitative analysis. Figure 3 shows an example (slightly modified for readability) of a code repair prediction and the span-copying actions. In this case, the model has learned to copy all of the input code in chunks, extending it only by inserting some new tokens in the middle.

Table 2: Span-based Correction (Bryant et al., 2017): Evaluation on Grammar Error Correction (GEC) Task. Note that our models use no pretraining, spell checking or other external data, which are commonly used in GEC tasks.

|  | Precision (%) | Recall (%) | $F_{0.5}$ |
|---|---|---|---|
| COPY+SEQ2SEQ | **34.9** | 6.4 | 0.1853 |
| SEQCOPYSPAN | 28.9 | **10.4** | **0.2134** |

For a quantitative analysis, we additionally compute statistics for the greedy decoding strategy of SEQCOPYSPAN.[2] In the BFP$_{small}$ dataset, SEQCOPYSPAN picks a Copy$(\cdot : \cdot)$ action with a span longer than one token about three times per example, copying spans eight tokens long on average (median six). This suggests that the model has learned to take advantage of the span-copying mechanism, substantially reducing the number of actions that the decoder needs to perform.

We also find that the COPY+SEQ2SEQ model tends to (mistakenly) assign higher scores to the input sequence, with the input sequence being predicted as an output more often compared to the span-copying model: the MRR of the input sentence is 0.74 for the baseline COPY+SEQ2SEQ model compared to 0.28 for the SEQCOPYSPAN model in the BFP$_{small}$ dataset. This suggests that the strong bias towards copying required of the baseline model (as most of the decoding actions are single-token copies) has the negative effect of sometimes "forgetting" to generate a change.

Finally, we use this task to evaluate our choices in the beam decoding algorithm Alg. 1 on the BFP$_{small}$ dataset (Table 1). First, using only a greedy decoding mechanism achieves an exact match 2.4% less often than our beam search method, indicating that beam decoding is still helpful. Second, when using a "standard" beam decoding algorithm in which we merge the probabilities of different rays only in a single post-processing step (*i.e.* directly implementing Eq. (3)), the accuracy of the top beam search result is only marginally worse, but the accuracy when considering the full beam is considerably affected. This is expected, as Alg. 1 ensures that rays are merged earlier, "freeing up" space in the beam for other results. This suggests that the added computational effort for merging beams allows the model to generate more diverse hypotheses. We also train and test model such that we provide supervision (teacher-force) copying the longest possible span, effectively disabling marginalization (Eq (2)), similar to Zhou et al. (2018). We find that the performance of the model significantly worsens (see Table 1). We believe that this is due to the fact that the model fails to capture the spectrum of correct actions possible since at each point it learns that only one action is correct.

**Grammar Error Correction** A counterpart to code repair in natural language processing is grammar error correction (GEC). Again, our span-copying model is a natural fit for this task. However, this is a rich area of research with highly optimised systems, employing a series of pretraining techniques, corpus filtering, deterministic spell-checkers, *etc.* We do not wish to compete with existing methods, but instead are only interested in showing that our method can also benefit systems in this setting. Instead we compare our SEQCOPYSPAN model to our baseline COPY+SEQ2SEQ model. Our models have a 2-layer bi-GRU encoder with a hidden size of 64, a single layer GRU decoder with hidden size of 64, tied embedding layer of size 64 and use a dropout rate of 0.2.

We use training/validation folds of the FCE (Yannakoudakis et al., 2011) and W&I+LOCNESS (Granger, 1998; Bryant et al., 2019) datasets for training and test on the test fold of the FCE dataset. These datasets contain sentences of non-native English students along with ground-truth grammar error corrections from native speakers. Table 2 shows the results computed with the ERRANT evaluation metric (Bryant et al., 2017), where we can observe that our span-copying decoder again outperforms the baseline decoder. Note that the results of both models are substantially below those of state of the art systems (*e.g.* Grundkiewicz et al. (2019)), which employ (a) deterministic spell checkers (b) extensive monolingual corpora for pre-training and (c) ensembling.

---

[2]We focus on greedy decoding here, as statistics in the presence of merged rays in the beam easily become confusing.

Table 3: Evaluation of models on the edit representation tasks of Yin et al. (2019).

|  | WikiAtomicEdits | GitHubEdits | C$^{\#}$ Fixers |
|---|---|---|---|
|  | Accuracy | Accuracy | Accuracy |
| Yin et al. (2019) | 72.9% | 59.6% | n/a |
| COPY+SEQ2SEQ | 67.8% | 64.4% | 18.8% |
| SEQCOPYSPAN | **78.1%** | **67.4%** | **24.2%** |

## 4.2 EDIT REPRESENTATIONS

We now turn our attention to the task of learning edit representations (Yin et al., 2019). The core idea of this task is to use an autoencoder-like model structure to learn useful representations of edits of natural language and source code. The model consists of an edit encoder $f_\Delta(x_-, x_+)$ to transform the edit between $x_-$ and $x_+$ into an edit representation. Then, a neural editor $\alpha(x_-, f_\Delta(x_-, x_+))$ uses $x_-$ and the edit representation to reconstruct $x_+$ as accurately as possible. We follow the same structure and employ our SEQCOPYSPAN decoder to model the neural editor $\alpha$. We perform our experiments on the datasets used by Yin et al. (2019).

Our editor models have a 2-layer biGRU encoder with hidden size of 64, a single layer GRU decoder with hidden size of 64, tied embedding layers with a hidden size of 64 and use a dropout rate of 0.2. In all cases the edit encoder $f_\Delta$ is a 2-layer biGRU with a hidden size of 64. The GRU decoders of both models use a Luong-style attention mechanism (Luong et al., 2015).

**Editing Wikipedia** First, we consider the task of learning edit representations of small edits to Wikipedia articles (Faruqui et al., 2018).[3] The dataset consists of small "atomic" edits on Wikipedia article without any special filters. Table 3 suggest that the span-copying model achieves a significantly better performance in predicting the exact edit, even though our (nominally comparable) COPY+SEQ2SEQ model performs worse than the model used by Yin et al. (2019). Our initial example in Figure 1 shows one edit example, where the model, given the input text and the edit representation vector, is able to generate the output by copying two long spans and generating only the inserted tokens. Note that the WikiAtomicEdits dataset is made up of only insertions and deletions and the edit shown in Figure 1 is generally representative of the other edits in the test set.

**Editing Code** We now focus on the code editing task of Yin et al. (2019) on the GitHubEdits dataset, constructed from small (less than 3 lines) code edits scraped from C$^{\#}$ GitHub C$^{\#}$ repositories. These include bug fixes, refactorings and other code changes. Again, the results in Table 3 suggest that our span-based models outperforms the baseline by predicting the edited code more accurately.

Yin et al. (2019) also use the edit representations for a one-shot learning-style task on a "C$^{\#}$ Fixers" dataset, which are small changes automatically constructed using automatic source code rewrites. Each edit is annotated with the used rewrite rule so that the dataset can be used to study how well an edit representation generalises from one sample to another.

As in Yin et al. (2019), we train the models on the larger and more general GitHubEdits dataset. To evaluate, we compute the edit representation $f_\Delta(x_-, x_+)$ of one sample of a group of semantically similar edits in C$^{\#}$ Fixers and feed it to the neural editor with the source code of another sample, *i.e.*, compute $\alpha(x'_-, f_\Delta(x_-, x_+))$. We repeat this experiment by picking the first 100 samples per fixer, computing the edit representation of each one and applying the edit to the other 99. The results of this process are shown in the last column of Table 3, suggesting that our span-copying models are able to improve on the one-shot transfer task as well.

Note that these results are not exactly comparable with those presented in Yin et al. (2019), as they randomly select 10 pairs $(x_-, x_+)$, compute their edit representation and then for a given $x'_-$ compute $\alpha(x'_-, f_\Delta(x_-, x_+))$ for each of the 10 edit representations, finally reporting the best accuracy score among the 10 candidates. Since this process cannot be replicated exactly due to the randomness of selecting samples, we opt for an alternate but reproducible process, as described above. Given, that

---

[3]According to Yin et al. (2019), a part of the data was corrupted and hence used a smaller portion of the data.

our SEQCOPYSPAN baseline is on par with the numbers reported in Yin et al. (2019) on the other tasks, we believe that our SEQCOPYSPAN improves the results. Table 4 in the appendix presents a breakdown of the performance on the fixer data per fixer, showing that for some fixers our model can substantially improve accuracy.

## 5 CONCLUSION

We have presented a span-copying mechanism for commonly used encoder-decoder models. In many real-life tasks, machine learning models are asked to edit a pre-existing input. Such models can take advantage of our proposed model. By correctly and efficiently marginalising over all possible span-copying actions we can encourage the model to learn to take a single span-copying action rather than multiple smaller per-token actions.

Of course, in order for a model to copy spans, it needs to be able to represent all possible plans which is $O(n^2)$ to the input size. Although this is memory-intensive, $O(n^2)$ representations are common in sequence processing models (*e.g.* in transformers). In the future, it would be interesting to investigate alternative span representation mechanisms. Additionally, directly optimising for the target metrics of each task (rather than negative log-likelihood) can further improve the results for each task.

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

Table 4: C# Fixer Accuracy (%) in the One-Shot Generation Task

| | COPY+SEQ2SEQ | | SEQCOPYSPAN | |
|---|---|---|---|---|
| | @ 1 | @ 5 | @ 1 | @ 5 |
| CA2007 | 16.8 | 24.4 | 36.9 | 46.5 |
| IDE0004 | 14.8 | 20.8 | 23.5 | 33.6 |
| RCS1015 | 24.0 | 25.3 | 23.9 | 26.8 |
| RCS1021 | 1.8 | 4.4 | 7.8 | 16.8 |
| RCS1032 | 1.8 | 2.7 | 2.5 | 3.7 |
| RCS1058 | 20.6 | 20.9 | 19.9 | 22.7 |
| RCS1077 | 3.2 | 3.9 | 4.5 | 5.8 |
| RCS1089 | 59.8 | 59.8 | 59.8 | 59.9 |
| RCS1097 | 1.6 | 3.7 | 14.9 | 27.7 |
| RCS1118 | 45.1 | 69.6 | 46.0 | 55.6 |
| RCS1123 | 15.8 | 19.5 | 27.7 | 22.7 |
| RCS1146 | 12.2 | 16.5 | 19.7 | 31.5 |
| RCS1197 | 1.1 | 1.8 | 1.7 | 2.3 |
| RCS1202 | 6.5 | 8.4 | 11.6 | 23.3 |
| RCS1206 | 34.9 | 35.0 | 36.2 | 37.5 |
| RCS1207 | 2.1 | 4.2 | 5.0 | 8.2 |

Table 5: Indicative evaluation of models on CNN/DM summarization.

| | BLEU | ROUGE-1 | ROUGE-2 | ROUGE-3 | ROUGE-L |
|---|---|---|---|---|---|
| COPY+SEQ2SEQ | 4.48 | 26.1 | 9.8 | 5.3 | 21.2 |
| SEQCOPYSPAN | **7.78** | **28.8** | **11.9** | **6.5** | **22.9** |

# A APPENDIX

## A.1 VISUALIZATION OF SPAN-COPYING ATTENTION

Figure 4 visualizes the copy-span attention for the greedy action sequence for the example in Figure 3.

## A.2 DETAILED FIXER EVALUATION RESULTS

Table 4 shows a breakdown of the performance of our models on the fixers dataset of Yin et al. (2019).

## A.3 RESULTS ON SEMI-EXTRACTIVE SUMMARIZATION

Abstractive and extractive summarization are two common tasks in NLP. Often abstractive summarization datasets, such as the CNN-Daily Mail corpus (Hermann et al., 2015) resemble extractive summarization to some extend. Here we aim to show that our SEQCOPYSPAN model can perform better than standard COPY+SEQ2SEQ models. However, given the time limitations, we do not fully replicate the summarization baselines and instead choose to use much smaller hidden states (each GRU has a hidden state of 64), fewer biRNN layers (2 encoder layers), smaller embedding size (of hidden dimension 64), a relatively small vocabulary (10k elements) *etc.* Our goal is merely to show how the two models compare.

Table 5 presents the results on commonly used evaluation metrics, showing that the SEQCOPYSPAN performs better than the simpler COPY+SEQ2SEQ. The examples in Figure 5 and Figure 6 show indicative summaries. Here the model learns to "copy-paste" full sentences or phrases to construct a summary. Note that in all these examples, the action sequence taken by our model is less than 5 actions.

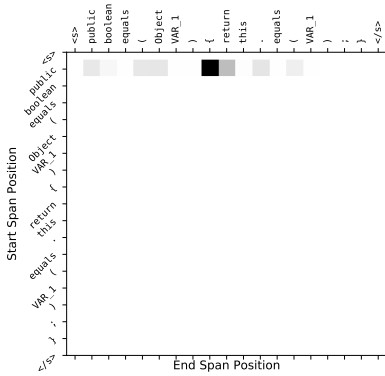

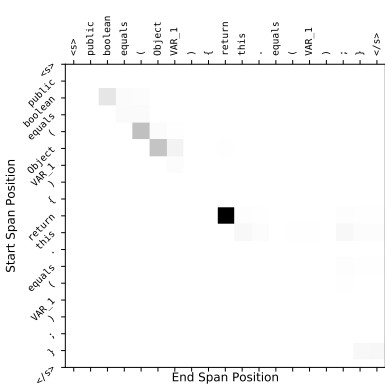

(a) $a_1$: Copy$(1:9)$ has a probability of 42.8%. The model is also predicting Gen($public$) with 0.4% probability.

(b) $a_2$: Gen(if) has probability 59.3% where as the highest span-copying action (Copy$(9:10)$) has a probability of only 0.8%.

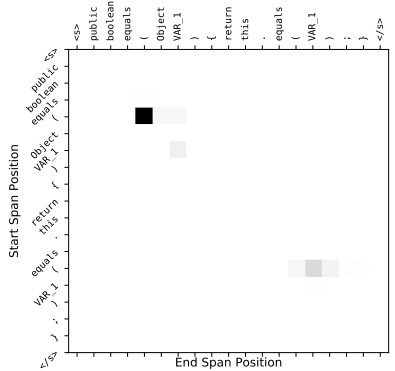

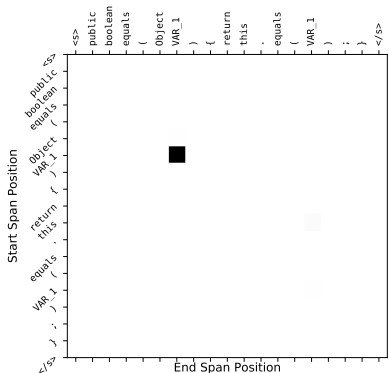

(c) $a_3$: Gen( ( ) has a probability of 99.9%. The highest span-copying action is the correct Copy$(4:5)$ but with negligible probability.

(d) $a_4$: Copy$(6:7)$ has a probability of 69.9%. The model is also (mistakenly) assigning a 24.3% probability to Gen( ( ).

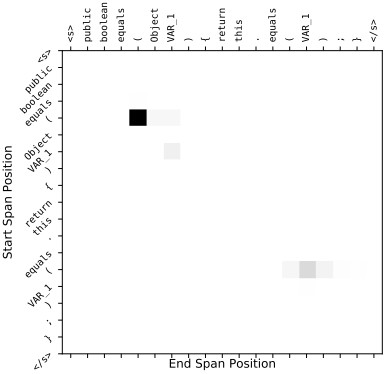

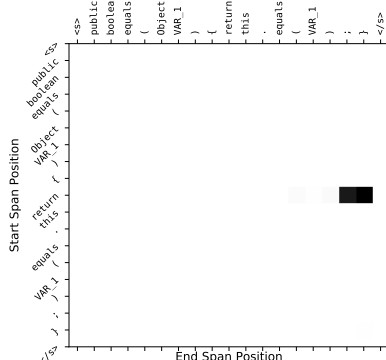

(e) $a_5$: Gen(==) has a probability of 66.0%. The highest span-copying action is Copy$(7:8)$ but with 0.8% probability.

(f) $a_8$: Copy$(9:18)$ has a probability of 43.7%. The model is also (mistakenly) assigning a 8.3% probability to Gen(else).

Figure 4: Visualization of the attention weights over some of the greedy action sequence of Figure 3. To fit this figure within one page, we only visualize attention at $a_1, a_2, a_3, a_4, a_5, a_8$. Note that the color range changes per-figure to allow for a better contrast in the visualization. Best viewed in screen.

**Text**

Scientists hoping to see 13 billion light years away , giving them a look into the early years of the universe, are facing opposition from Native Hawaiian groups looking to preserve their past. Demonstrators including Game of Thrones actor Jason Momoa demanded the state and University of Hawaii stop construction of a new $1.4 billion telescope on sacred land. Dozens of protesters were arrested on Thursday at the Mauna Kea site, a mountain burial ground said to be visited by the snow goddess Poli'Ahu and a Native Hawaiian leader has called for a 30-day moratorium on construction . Thirty-one people were arrested during protests blocking access to the construction site for the $1.4 billion Thirty Meter Telescope on Mauna Kea in Hawaii. Protesters say that the mountaintop , where scientists are building the facility to see 13billlion years into the past , is on top of sacred burial ground land . ...

**Predicted Summary:** Dozens of protesters were arrested at the Mauna Kea site in Hawaii.
**Actions:**

- Copy Span "Dozens of protesters were arrested"
- Copy Span "at the Mauna Kea site"
- Copy Span "in Hawaii".
- Generate `<eos>`

Figure 5: Sample 1: Semi-Extractive Summarization

**Text**

A Sydney teenage girl last seen leaving for school 40 years ago probably ran away and may still be alive , a coronial inquest has found. Marian Carole Rees was 13 when she disappeared from Hillsdale in southern Sydney in early April 1975 after telling a friend that she had forgotten something and jumped off her school bus. The teenager often talked of running away from home and had said goodbye to her brother on the morning she disappeared, Magistrate Sharon Freund said in findings handed down on Thursday. Marion Carole Rees -LRB- pictured -RRB- who went missing 40 years ago may still be alive, according to an inquest. ...

**Predicted Summary:** Marian Carole Rees went missing 40 years ago may still be alive.
**Actions:**

- Copy Span "Marian Carole Rees"
- Copy Span "went missing 40 years ago may still be alive"
- Generate ".".
- Generate `<eos>`

**Predicted Summary:** Marian Carole Rees was 13 when she disappeared from Hillsdale in southern Sydney in early April 1975
**Actions:**

- Copy Span "Marian Carole Rees was 13 when she disappeared from Hillsdale in southern Sydney in early April 1975"
- Generate ".".
- Generate `<eos>`

Figure 6: Sample 2: Semi-Extractive Summarization

## A.4 SPAN COPYING STATISTICS

In Figure 7, we plot the frequency of the lengths of the copied spans for $BFP_{small}$ and $BFP_{medium}$. Given that the merging mechanism in beam decoding does *not* offer a unique way for measuring the length of the copied spans (actions of different lengths are often merged), we disable beam merging for this experiments. Overall, the results suggest that the model learns to copy long sequences, although when single-copy actions are also common.

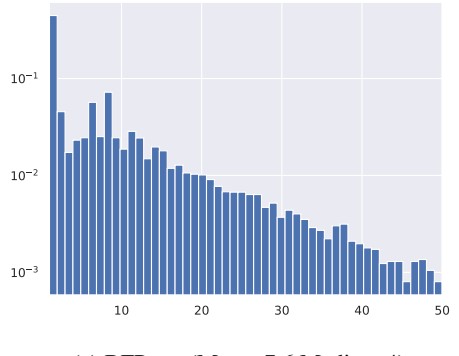

(a) BFP$_{small}$ (Mean: 7.6 Median: 4)

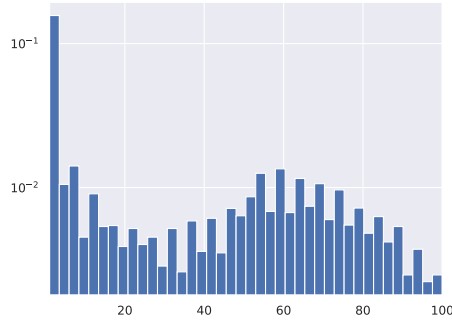

(b) BFP$_{medium}$ (Mean: 30.7 Median: 16)

Figure 7: Length histograms of Copy($\cdot$ : $\cdot$) actions during beam decoding in log-$y$ scale. Beam merging is disabled for computing the statistics of this experiment. For BFP$_{small}$ 43.7% of the copy actions are single-copy actions, whereas for BFP$_{medium}$ 37.6% of the actions are single-copy actions. This suggests that SEQCOPYSPAN uses long span-copying actions in the majority of the cases where it decides to take a span-copying action.

