# OpenReview forum: "Copy That! Editing Sequences by Copying Spans"
_ICLR.cc/2020/Conference — Reject_

### Official Review · AnonReviewer3 · 2019-10-22
**Official Blind Review #3**

**Rating:** 6

**Review:**

In this work, the authors tackle the problem of span-based copying in sequence-based neural models. In particular, they extend the standard copying techniques of  (Vinyals et. al., Gulcehre et. al., etc.) which only allow for single-token copy actions. Their span-based copy mechanism allows for multiple tokens to be copied at a time during decoding via a recursive formulation that defines the output sequence distribution as a marginal over the complete set of action combinations that result in the sequence being produced. The authors also propose a span-based beam decoding algorithm that scores output sequences via a sum over the probabilities of action sequences that produce the same output.

The authors evaluate their model on four tasks: code repair, grammar error correction, editing wikipedia, and editing code. They find that their proposed technique consistently outperforms single-token copy-based seq2seq baselines. They also show that the efficacy of their proposed beam decoding mechanism and do some simple quantitative analysis that the model learns to copy spans longer than a single token.


In general, I found this paper to be very clearly written with very good motivation for the proposed solution. In addition, I thought the authors did a good job of testing their model against a wide range of benchmark problems. It seems that their copy extension is a meaningful contribution.

I do, however, some questions regarding the evaluation, in particular the complexity of the baselines that were compared against. For example, the model consistently outperforms simple copy seq2seq baselines as well as the baselines in which the benchmark datasets were proposed (Tufano et. al, Yin et. al.) However, it does not seem the span-based copying method is state-of-the-art. If it is not state-of-the-art, how far off the SOTA is this proposed architecture? Did the authors do any analysis whereby the span-copy mechanism was added to an existing SOTA model, and if so, did this still produce gains? It's a bit difficult to situate the exact power of this new mechanism, given that it is often only compared to a simplistic copy-seq2seq method.

Other questions/feedback I have for the authors:
1) How efficient/scalable is the proposed mechanism? I would like to see a more formal treatment of the run-time of the training marginalization operation.
2) It would be nice to see a quantitative analysis for distribution of sequence lengths copied over (like some sort of histogram) for the datasets.
3) It would also be helpful to add some short descriptions of the benchmark datasets.



**Experience Assessment:**

I have published one or two papers in this area.

**Review Assessment: Checking Correctness Of Derivations And Theory:**

I assessed the sensibility of the derivations and theory.

**Review Assessment: Checking Correctness Of Experiments:**

I assessed the sensibility of the experiments.

**Review Assessment: Thoroughness In Paper Reading:**

I read the paper at least twice and used my best judgement in assessing the paper.

---

> ### Author Response · Authors · 2019-11-08
> **Response**
>
> Thank you for your comments and suggestions.
>
> •	Efficiency/Scalability: The computation cost of our algorithm during training is negligible. To identify all spans that can be copied from an input sequence of length $M$ in each location of a target sequence of length $N$ we use a standard dynamic programming approach [a] with cost $O(MN)$. This “alignment” can be pre-computed in the data pre-processing before training. During training, the marginalization has $O(N^2)$ cost (see Fig. 2: at each of the $O(N)$ decoding steps there are at most $O(N)$ possible lookbacks). In practice, this can be parallelized efficiently on a GPU, and given that the loss is a scalar, the main computational cost still rests on the vector/matrix operations rather than on the marginalization.
>
> At inference time, our beam search is not significantly slower than a more standard beam search (the only additional operation is the group_by_toks, which can be implemented using a HashMap in effectively constant time).
>
> •	We have added a histogram of the sequence length distributions in appendix A.4. More than 60% of the copy actions copy spans of length longer than a single token, in beam decoding on the BFP medium testset, with a median length of 16 tokens. Note that to measure this accurately, we disabled beam merging. Additionally, we added in Appendix A.1 visualizations of the span-copying attention for the example in Fig 3.
>
> •	We have added a description of the benchmark datasets when we introduce them in the text.
>
>
> [a] https://en.wikipedia.org/wiki/Longest_common_subsequence_problem

---

### Official Review · AnonReviewer2 · 2019-10-25
**Official Blind Review #2**

**Rating:** 3

**Review:**

This paper study the problem of editing sequences, such as natural language or code source, by copying large spans of the original sequence. A simple baseline solution to this problem is to learn a sequence to sequence neural network, which generates the edited sequence conditioned on the original one. This method can be improved by adding a copying mechanism, based on pointer networks, to copy tokens from the input. However, most of such existing approaches can only copy one input token at a time, which is a limitation when most of the input should be copied, which is the case for most editing tasks. In this paper, the authors propose a mechanism that can copy entire spans of the input instead of just individual tokens. In that case, a particular sequence can often be generated by many different actions (eg. copying individual tokens, pairs of tokens, or the whole span). It is thus important to marginalize over all the actions that generated a particular sequence. This can be done efficiently, using dynamic programming, if the probability of an action depends on the generated tokens only, but not on the sequence of actions used to generate them. In the case of neural network, this means that the decoder of the model takes the tokens as input, instead of the spans. To represent spans, the authors propose to use the concatenation of the hidden states corresponding to the beginning and end of the span. Then the probability of copying a span is obtained by taking the dot product between this representation and the current hidden state of the decoder, and applying the softmax. The authors evaluate the proposed approach on the following tasks: code repair, grammar error correction and edit representations (on wikipedia and c# code).

The paper is well written, and easy to follow, even if some sections could be a bit more detailed (for example, the section on
beam search decoding). The problem studied in the paper - copying spans from the input - is interesting, and has applications in NLP or code generation. I think that the the proposed solution is technically sound.
However, I have some concerns regarding the paper. First I believe that many relevant prior works are not discussed in the paper, making some technical contributions of the paper not novel. For example, previous methods were proposed to copy
spans from the input [1], to edit existing sequences [2], or to marginalize over different generation sequences
by conditioning only on the generated tokens (instead of the actions the generated the sequence) [3,4]. The body of work on iterative refinement for sequence generation is also probably relevant to this paper [5,6]. Additionally, I found the experimental section a bit weak, as most of the baseline used to compare seem a bit weak. The proposed method is mostly compared on datasets that are relatively new, or on tasks such as the grammar error correction where strong methods were excluded.

Overall, I found the paper well	written, and the proposed method to make sense. Unfortunately, I believe that the work is a bit incremental, most of the technical contributions having already been published. Since the experimental results are not very strong, I do not think the paper is good enough for publication to the ICLR conference.

== References ==

[1] Sequential Copying Networks, Qingyu Zhou, Nan Yang, Furu Wei, Ming Zhou, AAAI 2018.
[2] QuickEdit: Editing text & translations via simple delete actions, David Grangier, Michael Auli, 2017
[3] Latent Predictor Networks for Code Generation, Wang Ling, Edward Grefenstette, Karl Moritz Hermann, Tomas Kocisky, Andrew Senior, Fumin Wang, Phil Blunsom, ACL 2016
[4] Training Hybrid Language Models by Marginalizing over Segmentations, Edouard Grave, Sainbayar Sukhbaatar, Piotr Bojanowski, Armand Joulin, ACL 2019
[5] Mask-Predict: Parallel Decoding of Conditional Masked Language Models, Marjan Ghazvininejad, Omer Levy, Yinhan Liu, Luke Zettlemoyer, EMNLP 2019
[6] Deterministic Non-Autoregressive Neural Sequence Modeling by Iterative Refinement, EMNLP 2018

**Experience Assessment:**

I have published one or two papers in this area.

**Review Assessment: Checking Correctness Of Derivations And Theory:**

I assessed the sensibility of the derivations and theory.

**Review Assessment: Checking Correctness Of Experiments:**

I assessed the sensibility of the experiments.

**Review Assessment: Thoroughness In Paper Reading:**

I read the paper thoroughly.

---

> ### Author Response · Authors · 2019-11-08
> **Response**
>
> Thank you for your comments and suggestions. We would be glad to add more details to the paper as you are asking. Specifically, you mention the beam decoding section, could you let us know what information you’d have liked to see in that section that would make the algorithm and explanation more detailed and helpful?
>
> Thank you for bringing to our attention some of the work we have not cited. We have now cited them in the paper. However, we disagree that our work is too incremental for publication with respect to those works. Although some parts of our model are necessarily a combination of existing components, SpanCopy is different in significant ways from those works as follows:
>
> •	Sequential Copying Networks (Zhou et al. 2018) proposes to copy spans (for text summarization tasks) by predicting the start and end of a span to copy. However, it lacks marginalization over different actions generating the sequences and instead uses teacher forcing towards the longest copyable span; and it does not adapt its inference strategy to reflect this marginalization. This marginalization is the core contribution of our paper.
> As discussed in Sect. 2, using this marginalization was crucial for good experimental results. We have started running experiments comparing a variant without the marginalization and simple beam decoding in our final experimental setting and will report once these finished.
>
> •	QuickEdit (Grangier and Auli 2017) present a machine translation method that accepts a source sentence (e.g. in German) and a guessed translation sentence (e.g. in English) that is annotated (by humans) with change markers. It then aims to improve upon the guess by generating a better translations avoiding the marked tokens. This is markedly different as (a) the model accepts as input the spans that need to be removed or retained in the guess sentence. In contrast, CopySeqSpan needs to infer this information. (b) QuickEdit does mention having a copying mechanism neither for a single token nor for spans.
>
> •	Latent Predictor Networks (Ling et al. 2016) are one of the first papers to marginalize over a single-token copying action and a generation action yielding the same result. However, they do not consider copying spans of text, and hence only need to consider one decoding step at a time (essentially corresponding to the unnamed equation on page 2 of our submission).
>
> •	Hybrid LMs (Grave et al 2019), is similar to van Merriënboer et al. (2017), which we discuss in the related work section of our submission. These works also marginalize over different ways that yield the same output for language modeling (character-level, character n-gram and word-level). And although their training objective is similar to our objective in Eq. (2) they focus on language modeling without encoders, copying or span-copying, considering only a fixed set of possible outputs (words, character n-grams, characters). In contrast, here we consider arbitrary spans of the input and need to create representations of those spans.
>
> •	Ghazvinenejad et al (2019), Lee et al (2018) and the other non-autoregressive work we cited (Welleck et al., 2019; Stern et al., 2019; Gu et al., 2019) are indeed relevant but not directly related to this work as they can also be augmented with a span-copying mechanism.
>
> We have included the above in a revision of our submission.

---

> > ### Author Response · Authors · 2019-11-13
> > **Update**
> >
> > We retrained our model without using our new marginalized objective (Eq. 2), and instead forcing the model to copy the longest possible span at each timestep, as done in Zhou et al. (2018). We find that the performance of the model significantly worsens (see updated Table 1; “force copy longest”). We believe that this is due to the fact that the model fails to capture the spectrum of correct actions possible since at each point it learns that only one action is correct.

---

### Official Review · AnonReviewer1 · 2019-10-28
**Official Blind Review #1**

**Rating:** 6

**Review:**

This paper proposes a new decoding mechanism that allows  span-copying which can be viewed as a generalisation of
pointer networks.

Because action sequences of the same length can lead to sequences of different lengths decoding becomes tricky therefore authors propose a variation of standard beam search that calculates a lower bound of sequence probabilities rather than the true probability of generation this is achieved in practice by merging probabilities of sampled rays of actions generating the same sequence during the search.

one advantage of this proposed model is that it doesn't need to copy word by word to update sequences which need minor changes, rather than the seq2seq model with copy actions which due to the way we train those models using NLL loss will likely assign high probabilities to the non-modified input.

Authors evaluate their model against traditional seq2seq models with copy actions over a set of tasks:
* code correction tasks: two bug-fix pair (BFP) datasets of Tufano et al. (2019)
* grammar error correction (Bryant et al., 2017)
* learning edit representations

Pros:
Overall I am in favour of this work acceptance it represents a neat modelling for copying sequences that integrated simply with seq2seq models, especially the transformer model.

Cons:
- One of the drawbacks of this method is the decoding strategy although authors present a motivated solution for that. The proposed variation of beam search by calculating the lower bound solution seems adhoc and some corner cases are not explained in the paper (see the question below).
- Experiments could have been more thorough, especially in terms of architectures. I was disappointed not to see authors only comparing between GRU based seq2seq with copy actions, one baseline and their model implementation over a biGRU seq2seq.

Questions to authors:
- During inference using the proposed variation of beam search (e.g. k=5), What will happen for example if one ray of actions was dropped because not of the top 5 this ray of actions if continued using future actions would map to one existing top-scoring rays? do you do a way to control that?
- What is the reason behind choosing the bi-gru architecture?

Missing references:
There are a couple of similar work that authors might want to add:
* Latent Predictor Networks for Code Generation https://arxiv.org/pdf/1603.06744.pdf
* An Operation Network for Abstractive Sentence Compression https://www.aclweb.org/anthology/C18-1091.pdf
* Levenshtein Transformer https://arxiv.org/pdf/1905.11006.pdf

**Experience Assessment:**

I have published one or two papers in this area.

**Review Assessment: Checking Correctness Of Derivations And Theory:**

I carefully checked the derivations and theory.

**Review Assessment: Checking Correctness Of Experiments:**

I carefully checked the experiments.

**Review Assessment: Thoroughness In Paper Reading:**

I read the paper thoroughly.

---

> ### Author Response · Authors · 2019-11-08
> **Response**
>
> Thank you for your comments and suggestions. To answer your questions:
>
> •	Re baselines: Apologies for the imprecision in our text: our baseline is a seq2seq model where the encoder is a (two-layer) biGRU and the decoder is a GRU. More precisely, all of our baselines are using the same codebase with span-copying turned off. We have made this explicit in the introduction of the evaluation section.
>
> •	Re biGRU architecture: The choice of (bidirectional) GRU networks was motivated by cursory early experiments in which GRU models performed similarly well as LSTM models, but were slightly faster.
>
> •	Re beam search: We are not sure we fully understand the question. The problem you seem to describe (that a low-scoring ray could eventually be completed to a high-scoring ray, but is not explored if it falls out of the beam) is a general problem of beam search, and does not seem to be influenced by our modification. The prevalence of this problem for a given decoder can be approximated by running beam search with substantially bigger beams and comparing the results (where the “gap” stems from rays that were dropped). We are running such an experiment on BFP small with a beam size of 100 and will report these results when they are finished.
>
> •	Re corner cases of beam search: We did not understand which corner cases are unclear. Could you provide more details on this?
>
> •	Re related work: We have added a comparison to “Latent Predictor Networks for Code Generation” in our latest revision, and were already citing the Levenshtein Transformer in our discussion of decoders that are not following a left-to-right decoding strategy.

---

> > ### Author Response · Authors · 2019-11-13
> > **Update**
> >
> > We ran the evaluation with a beam size of 100 (See Review #1). With a beam size of 100 the results on BFP small are as follows:
> >
> > •	Exact Match: 17.89% (+0.2 increase from using a beam size of 20).
> >
> > •	MRR: 0.250 (from 0.247 when the beam size is 20)
> >
> > Overall, these suggest that the improvements are minimal. Using an alternative beam search method, that would preserve (e.g.) beams that contain a lot of copied tokens (at the cost of memory) does not seem to provide any particular advantage.

---

### Author Response · Authors · 2019-11-08
**General Review Response and Paper Update (8 Nov)**

Thank you for all your thoughtful feedback and questions. We want to clarify some general points, and document the changes to a revision to the paper we have uploaded now:

•	Experimental Results: Our model achieves new state-of-the-art results on the considered code-related tasks as well as on the natural language edit representation task. For the grammar error correction task, we compare the model to a baseline model similar to the one used in state-of-the-art works, but do not perform any of the pre-processing and pre-training steps, since these aspects are not a core novelty of our approach.

•	Related Work: We have extended our related work section with some of the works pointed out by the reviewers. This includes references to “Latent Predictor Networks for Code Generation” (Ling et al. 2016), which to our knowledge is the first to marginalize over copying actions and generation actions yielding the same results. Our is a generalization of this marginalization strategy to spans of text.

We now also compare to “Sequential Copying Networks” (Zhou et al. 2018), which also provides a mechanism for copying subsequences of the input, but does not present marginalization strategies over action sequences generating the same result, which are the main contribution of our paper.

•	Additional Visualizations: We have added Appendix A.1 and A.5 with additional visualization about the span-copying mechanisms. This should hopefully provide better insights on the inner workings of our method.

---

### Decision · Program_Chairs · 2019-12-19

**Decision:**

Reject

**Comment:**

This paper proposes an addition to seq2seq models to allow the model to copy spans of tokens of arbitrary length in one step. The authors argue that this method is useful in editing applications where long spans of the output sequence will be exact copies of the input. Reviewers agreed that the problem is interesting and the solution technically sound. However, during the discussion phase there were concerns that the method was too incremental to warrant publication at ICLR. The work would be strengthened with a more thorough discussion of related work and additional experiments comparing with the relevant baselines as suggested by Reviewer 2.